# Determining an Optimal Oxygen Saturation Target Range Based on Neonatal Maturity: Demonstration of a Decision Tree Analytic

**DOI:** 10.3390/diagnostics13213312

**Published:** 2023-10-26

**Authors:** Thomas E. Bachman, Narayan P. Iyer, Christopher J. L. Newth, Robert LeMoyne

**Affiliations:** 1Faculty of Biomedical Engineering, Czech Technical University in Prague, 272 01 Kladno, Czech Republic; lemoyrob@fbmi.cvut.cz; 2Children’s Hospital Los Angeles, University of Southern California Keck School of Medicine, Los Angeles, CA 90027, USA; niyer@chla.usc.edu (N.P.I.); cnewth@chla.usc.edu (C.J.L.N.)

**Keywords:** machine learning, decision tree classification, oxygen saturation targeting, neonatal, hyperoxemic-risk, hypoxemic-risk

## Abstract

The utility of decision tree machine learning in exploring the interactions among the SpO_2_ target range, neonatal maturity, and oxemic-risk is demonstrated. METHODS: This observational study used 3 years of paired age-SpO_2_-PaO_2_ data from a neonatal ICU. The CHAID decision tree method was used to explore the interaction of postmenstrual age (PMA) on the risk of extreme arterial oxygen levels at six different potential SpO_2_ target ranges (88–92%, 89–93%, 90–94%, 91–95%, 92–96% and 93–97%). Risk was calculated using a severity-weighted average of arterial oxygen outside the normal range for neonates (50–80 mmHg). RESULTS: In total, 7500 paired data points within the potential target range envelope were analyzed. The two lowest target ranges were associated with the highest risk, and the ranges of 91–95% and 92–96% were associated with the lowest risk. There were shifts in the risk associated with PMA. All the target ranges showed the lowest risk at ≥42 weeks PMA. The lowest risk for preterm infants was within a target range of 92–96% with a PMA of ≤34 weeks. CONCLUSIONS: This study demonstrates the utility of decision tree analytics. These results suggest that SpO_2_ target ranges that are different from typical range might reduce morbidity and mortality. Further research, including prospective randomized trials, is warranted.

## 1. Background

Preterm infants spend many weeks after birth in the neonatal intensive care unit (NICU). As a consequence of the instability of their underdeveloped respiratory system, supplemental oxygen and ventilatory support are usually required for prolonged periods. Further, their immaturity also makes them highly susceptible to excess mortality and morbidity. Pulmonary, retinal and neurodevelopmental morbidity are specifically associated with not only inadequate oxygenation but also excessive oxygen [1,2,3].

Due to their respiratory instability, managing neonatal oxygenation is challenging, requiring frequent adjustments to the inspired oxygen [4]. While intermittent arterial blood gas measurements are feasible, with limited arterial access, they are not an acceptable tool for monitoring oxygenation in unstable neonates. Rather, arterial sampling is used primarily for periodic assessments of the acid–base balance. The standard of care for managing oxygenation in neonates is to obtain continuous noninvasive peripheral oxygen saturation (SpO_2_) measurements [5,6]. While SpO_2_ is continuous, it is not as accurate as arterial measurements and is relatively insensitive to changes in higher levels of arterial oxygen (PaO_2_) [7]. Because these infants are highly susceptible to both high (hyperoxemia) and low (hypoxemia) oxygen, SpO_2_ is managed in a tight target range. Currently, the usual SpO_2_ target range in the neonatal ICU is 90–95% [8,9,10,11]. However, European guidelines recommend 90–94% for extremely preterm infants [5]. Finally, the target range has changed dramatically in the last several decades. Based on pragmatic outcome studies, it has shifted from previously higher norms (95–98%) [12] to rather lower levels (85–93%) [13,14]. While it is now between 90 and 95%, this range is far from a consensus. Some think the optimum range should be either slightly lower [15] or slightly higher [16,17].

Most also believe that the maturity of the infant should be a factor in determining the target range, with slightly higher oxygen levels being the norm. The traditional transition from prematurity is above 36 weeks postmenstrual age (PMA). Maturity is a factor for two reasons. The first is that excess oxygenation in significantly premature infants has long been associated with blindness [3]. While this is now rare, retinal morbidity remains a common problem. However, increased exposure to intermittent low oxygenation has been shown to increase mortality [2] and has even been associated with retinal morbidity [18]. Secondly, the relationship between arterial oxygenation (PaO_2_) and oxygen saturation varies in preterm infants as they transition from fetal to adult hemoglobin. This gestational age hemoglobin transition effect has its primary impact via the increased risk of high arterial PaO_2_ associated with adult hemoglobin. Clinically, relevant differences in the risk of high oxygen between preterm and near-term infants have been demonstrated. [19].

According to standard nomenclature, infants born between 37 and 39 weeks are characterized as late preterm, those born ≤36 weeks gestation are characterized as preterm and those born ≥40 weeks are characterized as term [20]. To normalize for differences in gestational and postnatal ages, PMA is commonly used in neonatology for the assessment of outcomes. It is determined by adding the gestational age to the postnatal age. The assessment of neonatal outcomes, to normalize for gestational age, is based on status at 36 weeks PMA. Thus, an infant born at 24 weeks gestational age would be assessed when they are 12 weeks old, while an infant born at 35 weeks would be assessed when they are one week old. The use of 36 weeks PMA is based on the premise that an infant’s physiological development continues in the same trajectory after birth as during intrauterine maturation. However, applying this traditional outcome assessment criteria to the selection of maturity-based SpO_2_ targets might not be appropriate.

The roots of machine learning or artificial intelligence were established over half a century ago. The exponential growth of computer processing power, coupled with the development of alternative analytical approaches, has resulted in it becoming embedded into our daily lives. Its use in healthcare has become essential for administrative tasks such as optimizing scheduling and resource management. Further, the opportunities for computer-enhanced medical devices and diagnostic/therapeutic decision support are apparent. Originating from decision support practice, automated decision trees have become common tools in machine learning [21]. Given a data set, they parcel it into distinctly different subsets called leaves. Different decision tree methods use distinct approaches. However, the simplicity of the approach and the comprehensible nature of the results are common attractive traits. This is in contrast to the “black box” methodology of other machine learning approaches, such as fuzzy logic, random forests and neural networks.

The aim of this study was to demonstrate how a machine learning decision tree analysis could be used to determine optimal cutoffs for maturity to minimize the risk of inappropriate oxygenation in different potential target SpO_2_ ranges.

## 2. Methods

### 2.1. Design

This was an observational study using data from a data repository at a single NICU. The collection and use of the data were approved by the requisite hospital research ethics committee. The need for informed consent was waived as there was no research intervention and the patient identities are masked.

### 2.2. Population

The Children’s Hospital Los Angeles, University of Southern California Keck School of Medicine, is a major regional tertiary care referral center. Its neonatal services include a 58-bed NICU. The hospital does not include an obstetrics service, and admissions to the level 4 NICU are referred from other NICUs for a higher level of care. The patient data repository represents 3 years of experience of neonates receiving supplemental oxygen and mechanical ventilatory support. It includes 26,162 paired PMA-SpO_2_-PaO_2_ measurements.

### 2.3. Data Collection

Details of the collection of the requisite study data from the patient data repository were provided previously described. [19]. In summary, the query strategy included several steps. First, an average SpO_2_ value was calculated for each arterial blood gas measurement. SpO_2_ data from the monitoring system were stored every 30 s. The average value was the mean of all values within 120 s of the arterial sample. Second, the gestational age was accessed for each neonate. The PMA was then calculated as the sum of the birth gestational age and the postnatal age at the time of the arterial sample.

### 2.4. Endpoints

For this evaluation, six potential SpO_2_ target ranges were defined that spanned the range of most practices [8,9,10,11]. These ranges were centered at six median SpO_2_ levels (90, 91, 92, 93, 94, 95%) with a span of ±2% SpO_2_. While target ranges often have a span wider than ±2, this narrower span was selected to facilitate the detection of subtle differences across the possible target ranges. Thus, the SpO_2_ ranges evaluated were 88–92%, 89–93%, 90–94%, 91–95% 92–96% and 93–97%. Maturity was defined as the estimated gestational age at birth plus the postnatal age (PMA) and was described in weeks and days (week^days^). The paired PMA-SpO_2_-PaO_2_ in the database reflect when blood gas measurements were ordered, the timing of which might be associated with a standing order, a clinical exacerbation or an assessment of the effectiveness of weaning from respiratory support. Thus, it was not a random sample of oxygenation when stable. Therefore, 750 paired measurements were randomly selected from each of the 10 SpO_2_ bins (88–97%).

We labeled the primary dependent variable oxemic-risk, which was quantified using a risk score assigned to each measurement. The score was based on a risk-weighted average of measurements of PaO_2_ outside the normal range for neonates (50–80 mm of Hg). The weights were based on a subjective estimate. More relative weight was placed on hypoxemia than on hyperoxemia, based on recent outcome studies [1,2]. These studies found that lower levels of SpO_2_ were associated with increased mortality, while exposure to higher levels was associated with poorer developmental outcomes. Significant levels of hypoxemia (PaO_2_ ≤ 40 mmHg) were scored as 6, and those between 41 and 49 mmHg were scored as 2. Oxemic-risk associated with hyperoxemia was scored as follows: ≥200 mmHg as 6, 100–199 mmHg as 3 and 81–99 mmHg as 1.

### 2.5. Statistical Considerations

The primary aim of this analysis was to determine the PMA thresholds, if any, at which the oxemic-risk shifted for each of the six different potential target ranges. The CHAID decision tree method was used to determine these PMA break points [22]. CHAID (Chi-squared Automatic Interaction Detector) was selected from other decision tree methods for two reasons. First, other decision tree machine learning analytics are binary and multitiered and thus provide hierarchical output. The CHAID analysis is nonbinary. Thus, it can identify the number of significantly different subgroups, whether there are zero or many. Further, it uses a traditional chi-squared method to partition the data into the relevant number of statistically significant categories. This avoids the lack of confidence associated with some machine learning approaches that utilize unintelligible or “black box” analysis methods. For illustrative purposes, Figure 1 shows the decision tree for oxemic-risk for all 21,162 paired measurements and was created using CHAID. In this case, Four unique PMA partitions were identified. The number of measurements, their proportion and the oxemic-risk are shown for each. As an example, for the lowest of the four PMA partitions (≤37 weeks), there were 5256 measurements (20.1%), with an oxemic-risk of 1.87. The oxemic-risk values for the four partitions (1.87, 2.45, 2.67 and 2.26) are statistically significantly different.

Populations included all data points regardless of SpO_2_. PMA is in weeks^days^.

Subsequent to the primary analysis, two descriptive analyses were conducted. The first post hoc analysis was calculation of the oxemic-risk score for a common recommendation (90–94% SpO_2_ target range for PMA partitions ≤36 weeks) and contrast it with those with >36 weeks PMA. The second was a calculation of the oxemic-risk for those SpO_2_ measurement outside the target range envelop. These were hypoxemic SpO_2_ measurements (<88%) and hyperoxemic SpO_2_ measurements (>97%).

Demographics of the neonates, and a variety of cross tabulations of SpO_2_ and PaO_2_ were compiled. The CHAID-derived partitions were, by definition, statistically significantly different. Confidence limits of the descriptive oxemic-risk scores were calculated, with statistical significance inferred if they did not overlap. The CHAID and other analyses were conducted using XLSTAT v11.5 (Lumivero, New York, NY, USA).

## 3. Results

In the analysis, there were 3750 paired data points in each of the six specified target ranges and 7500 in the envelope of normoxemia between 88 and 97% SpO_2_. These data came from 888 infants with a median gestational age of 37 weeks (IQR 32–39) at birth. The median number of samples per subject was five (IQR 2–10). The median postnatal age of the infants at the time of the blood gas measurements was 2.0 weeks (IQR 0.9–4.9). The PMA, SpO_2_ and PaO_2_ levels for these neonates are shown in Table 1. For comparison, the corresponding values above and below the SpO_2_ target range envelope (88–97%) are also shown. The PMA values were comparable, and in confirmation of the expectation, the oxygenations levels are markedly different (SpO_2_: 94, 80 and 99%, and PaO_2_: 42, 74 and 129 mmHg).

The PaO_2_ and SpO_2_ values for the measurements in the target range (SpO_2_ 88–97%) for each of the six PaO_2_ oxemic-risk strata are provided in Table 2. The PaO_2_ levels are consistent with the six oxygenation strata, increasing from a low of 37 ± 4 to a high of 273 ± 64 mmHg. In contrast, while the SpO_2_ increased by one mmHg for each of the two steps of hypoxemia, it is insensitive to increases in PaO_2_ between the three higher PaO_2_ risk strata that are associated with hyperoxemia. It is also of note that nearly half (42%) of the PaO_2_ readings associated with SpO_2_ readings between 87 and 97%, are outside the normal of PaO_2_ range of 50–80 mmHg. This is reflective of a known limitation in the precision of SpO_2_ measurements.

Table 3 describes the SpO_2,_ PaO_2_ and associated oxemic-risk scores for the six SpO_2_ target ranges. The PaO_2_ increases from 56 ± 24 mmHg for the target range of 88–92% SpO_2_, with increasingly higher SpO_2_ target ranges of up to 80 ± 41 mmHg for the highest target range (93–97% SpO_2_). The large standard deviations of the PaO_2_ in each target range also reflect the lack of precision in the SpO_2_-PaO_2_ relationship. These oxemic-risk scores are “U shaped”. The highest (1.24) is associated with the lowest SpO_2_ target range (88–93%). SpO_2_ target ranges of 91–95% (0.83) and 92–96% (0.80) are at the nadir, and the 95% confidence limits of their oxemic-risk scores indicate that they are not different. The highest SpO_2_ target range (93–97%), just 1% SpO_2_ higher, has a higher oxemic-risk score of 0.88.

Table 4 summarizes the ideal PMA oxemic-risk partitions derived from the CHAID decision tree for each SpO_2_ target range. For all six target ranges the upper partition cut off was during the 42nd week of PMA. The oxemic-risk for these older neonates was lower at each SpO_2_ target range than the oxemic-risk for the less mature infants. Three of these PMA partitions were dichotomous (89–93%, 90–94% and 91–95%), that is there was no significant additional partitions below 42 weeks. An additional partition was identified for the other 3 target ranges, though at different PMA thresholds (38, 39, 34 weeks). The lowest oxemic-risk for immature infants was with the 92–96% target range and ≤34 weeks PMA. In addition, the oxemic-risk was markedly higher in the two lower target ranges (88–92%, 89–93%) than the other 4 target ranges for each PMA partition. Interestingly the oxemic-risk for the middle PMA partition, when present, is higher than the low and high PMA partitions.

We also determined the oxemic-risk for the recommended target range of 90–94%, above and below the usual transition at 36 weeks (≤36 weeks 0.97 (95% CI 0.89–1.06), >36 weeks 0.94 (0.88–1.00)). These were not statistically different and were also markedly higher risks than the optimal target ranges identified via the decision tree analysis. Shown graphically in Figure 2 are these oxemic-risk scores and PMA partitions, along with the optimal 92–96% target range, as well as those for SpO_2_ < 88% and SpO_2_ > 97%. The oxemic-risk when the SpO_2_ is not in the target range is nearly an order of magnitude larger than when it is in the target range. The difference between the two target ranges for PMA ≤ 34 and ≥42 is also marked.

## 4. Discussion

We used a machine learning decision tree to evaluate the complex interaction of two parameters: infant maturity and six different potential SpO_2_ target ranges. The goal was to determine the best target range for infants of differing maturities based on minimizing oxemic-risk. Our results suggest that current practice of using a target range of 90–95% and transitioning to higher levels at >36 weeks might not be ideal for minimizing the risk associated with excessive and inadequate oxygen.

We found that the lowest oxemic-risk occurred in a target range of 92–96% SpO_2_ but, as expected, the risk varied depending on the neonate’s maturity. The lowest oxemic-risk occurred in term infants (PMA ≥ 42 weeks) regardless of the target range. Among these mature term infants, a target range of 91–95% resulted in a reduced oxemic-risk ranging between 5 and 72% compared to the other five target ranges. We believe this target range is lower than in typical practice. Though little has been published to support our perception, the finding is consistent with another analysis. [19] For extremely preterm infants (PMA ≤ 34 weeks), a target range of 92–96% was associated with a reduced oxemic-risk ranging between 11 and 70% compared to the other five target ranges. Specifically, it was a 55% reduction from the European recommendation of 90–94% [5]. This 34-week cut off was also two weeks less than the standard practice of ≥36 weeks. For those infants between 35–41 weeks PMA, the oxemic-risk was the lowest for the target ranges of 91–95% and 92–96%.

In summary, the more common practice is to use a target range of 90–95% for preterm infants [8,9,10], and some advocate using lower targets until 36 weeks PMA [13,14]. In contrast, our analysis of oxemic-risk suggests a target range of 92–96% is better, particularly for extremely preterm infants. Nevertheless, this was a focused analysis and only one part of the puzzle. Other factors also ought to be considered [6]. One of these factors is the percent of hemoglobin that is fetal. Another is the ability of the clinical staff to be compliant with the intended target range. Several other factors beyond the scope of this discussion are body temperature, congenital anomalies, the severity of pulmonary dysfunction, and the total hemoglobin level. For this reason, the recommendation of the American Academy of Pediatrics suggests that targeting should be more patient-specific rather than uniform [6].

The relationship between arterial oxygen saturation and oxygen partial pressure is complex. First the relationship shifts according to the concentration of fetal hemoglobin. Thus, the concentration of fetal hemoglobin, associated with neonatal development, results in a need for maturity-based target ranges. However, the predicable transition over several months shifts with blood transfusions. In addition, the acid–base balance, temperature and other factors also result in clinically relevant shifts. Second, the SpO_2_ is a noninvasive measurement of arterial oxygen saturation (SaO_2_). These two factors account for much of the scatter between the predicted PaO_2_ and the measured SpO_2_. Finally, the relationship is not linear but rather sigmoid, and the clinically relevant part, above 50% saturation, has been described by a seventh order polynomial [23]. This sigmoid relationship explains the poor sensitivity of high SpO_2_ levels to increasingly higher PaO_2_s above the target range as the relationships becomes asymptotic with 100% SpO_2_. In contrast, there is linear drop in PaO_2_ and SpO_2_ values below the target range, making it reliable for detecting hypoxemia, even if the exact value is not precise. Nevertheless, SpO_2_ is the standard of care for monitoring oxygenation. When considering trade-offs between target ranges it is also highly relevant that lower SpO_2_ target ranges, as the result of a linear drop in the PaO_2_, have been shown to limit the oxygenation reserve of the infant and thus result in more frequent episodes of hypoxemia [24,25]. Therefore, lower target ranges, while reducing the risk of hyperoxemia with a lower average PaO_2_, also increase the prevalence of hypoxemic episodes. This consideration provides additional support to our findings that higher target ranges for less mature neonates would reduce oxemic-risk.

Until recently, oxygen levels in the NICU were managed exclusively by manual transient increases in the inspired oxygen in response to desaturations. This approach has been shown to result in only about half of the time being spent in the desired SpO_2_ target range, with an additional marked bias between hypoxemia and hyperoxemia among centers and caregivers [4,11]. Of late, many new neonatal ventilators offer closed-loop control over inspired oxygen based on SpO_2_. This approach to titrating inspired oxygen has consistently been shown to improve time spent in the intended target range and to reduced time spent at SpO_2_ extremes [26]. Aside from enhanced general effectiveness, these automatic control systems not only make the use of narrower target ranges practical but also offer the possibility of regular changes to target ranges that are individualized for each neonate.

The design of this study and its results are somewhat unique. This is, to our knowledge, the first study to use a severity-weighted risk score in addition to one that balances the risk between hypoxemia and hyperoxemia. In addition, only one other study has examined the exposure to aberrant arterial PaO_2_ levels when within the intended target range [19]. Studies of neonatal SpO_2_ targeting have primarily investigated maintaining compliance with the SpO_2_ target and the amount of time spent outside a target range and at SpO_2_ extremes [26]. Our results focus on this balanced tradeoff of oxemic-risk when in the target range and also provide a contrast to situations in which saturations are above or below normoxemia. The unique use of machine learning also suggests that traditional transitions in risk, based on maturity, ought to be reexamined. This approach also identified an interim maturity level with elevated risk. While it is well understood that preterm neonates experience a period of increased instability when they are 2–4 weeks old, which clearly affects the ability to maintain normoxemia, our finding suggests that even in the target range, they have an increased oxemic-risk.

Another similar analysis which only considered the maintenance of PaO_2_ between 50 and 80 mmHg found no difference in similar ranges and in preterm vs. near term infants. [19] Our analysis considered the severity of the risk of values outside the desired PaO_2_ range rather than the prevalence within. Our analysis used one scheme to assign weights to different levels of hyperoxemia and hypoxemia for all infants. Though defendable, it can be challenged as perhaps overly simplistic. Nevertheless, it is not clear what a better weighting strategy might be. Some experts argue that extremely preterm infants have no mechanisms to protect themselves from excessive oxygenation and that prior to delivery, their fetal oxygen was <40 mmHg, and they are thus are at added risk [14,15]. In contrast others suggest that fetal hemoglobin limits the likelihood of high arterial oxygenation and thus is protective [16,19].

Of note, the data used in this study came from a level 4 tertiary care center. This permitted an analysis of the experience of neonates on extended respiratory support over a three-year period that might only be treated over decades in most of the referring NICUs. We believe it is generally applicable to other neonates receiving respiratory support, albeit for shorter periods of time. It is not generally applicable to neonates just receiving supplement oxygen. However, consideration should be given to the increased likelihood that the neonates in our study population received one or more transfusions of adult blood and thus their shift to adult hemoglobin was accelerated. Nevertheless, we would expect that this would increase exposure to hyperoxemia in our population and thus probably reinforce our overall finding that higher ranges for preterm infants might be preferable. However, it might well change the relative oxemic-risk scores and the PMA transition points that our study identified.

In addition to the issues discussed above, our study has other limitations. First, this was an observational study, the results of which should always be considered with caution. Second, the level of inspired oxygen might also play a role in the risk of hyperoxemia and was not considered. Severe PaO_2_ levels are much less likely, though possible, when inspired oxygen is weaned towards room air. However, most hyperoxemia in infants is a result of administering excess oxygen following a period of hypoxemia, which can happen at any baseline inspired oxygen level [27]. Finally, the analysis was based on a traditional normal PaO_2_ range of 50–80 mmHg. This range has not been, and never will be, carefully correlated to normal outcomes. Rather, SpO_2_ monitoring is the standard of care, and studies are needed to explore the impact of SpO_2_ target ranges higher than 90–94% or 90–95%.

Our findings have some important implications related to needed research. Further prospectively designed observational studies in other subject populations are needed. With the increasing number of centers collecting and archiving data from integrated patient-monitoring systems, this is practical and efficient. Specific study populations should include extremely preterm neonates in the early stage following birth as a well as neonates who have not been referred to tertiary centers for a higher level of care. Further analyses of the sensitivity of changes in oxemic-risk weighting as well as oxemic-risk scores that change with maturity are warranted. These results would contribute to the design of randomized controlled multicenter trials essential to supporting rationale recommendations for SpO_2_ target ranges not just for extremely preterm neonates but for all neonates requiring respiratory support.

## 5. Conclusions

Our study demonstrated the usefulness of an automated decision tree to quickly partition data into statistically different groups, independent of historical rubrics. It was found to be an effective analytical tool. Our results also suggest that SpO_2_ target ranges slightly higher than commonly used might have benefit for preterm infants and slightly lower for near term and term infants. Further prospective studies in other subject populations are warranted.

## Figures and Tables

**Figure 1 diagnostics-13-03312-f001:**
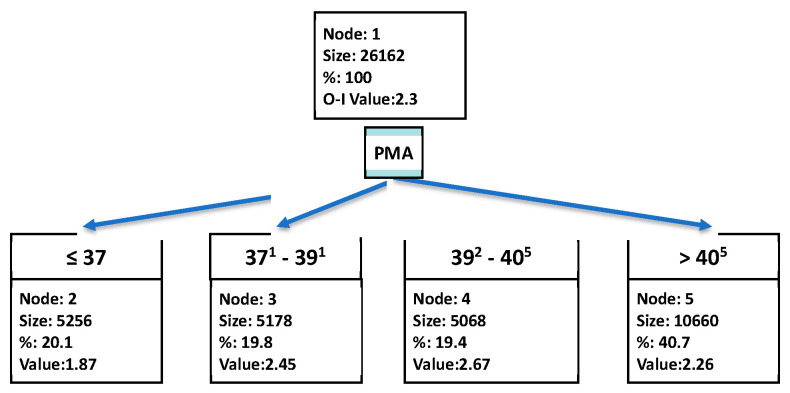
Decision tree for oxemic-risk based on PMA.

**Figure 2 diagnostics-13-03312-f002:**
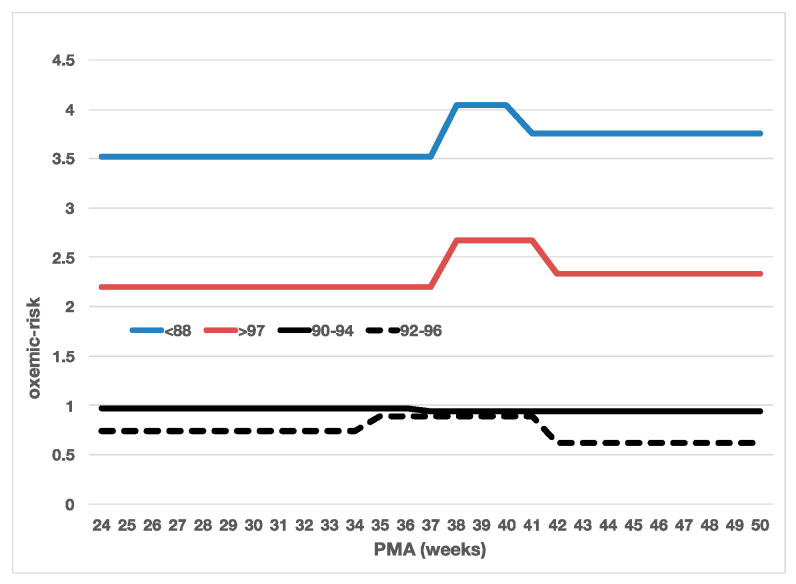
Comparison of oxemic-risk within and outside target ranges.

**Table 1 diagnostics-13-03312-t001:** PMA, PaO_2_ and SpO_2_ in, below and above the target ranges.

SpO_2_ Strata	*n*	PMA (Weeks)	PaO_2_ (mmHg)	SpO_2_ (%)
88–97%	7500	40 ± 7	74 ± 40	94 ± 3
<88%	6667	41 ± 8	42 ± 16	80 ± 4
>97%	10,652	40 ± 7	129 ± 66	99 ± 1

Means ± STDs.

**Table 2 diagnostics-13-03312-t002:** PaO_2_ and SpO_2_ in each PaO_2_ Strata for 88–97% SpO_2_.

PaO_2_ Strata	*n*	%	SpO_2_	PaO_2_	Oxemic-Risk
≤40	381	5.1%	90 ± 2	37 ± 4	6
41–49	1398	19%	91 ± 2	46 ± 2	2
50–80	4367	58%	93 ± 3	62 ± 8	0
81–99	731	9.7%	95 ± 2	88 ± 5	1
100–199	501	6.7%	95 ± 2	127 ± 27	3
≥200	122	1.6%	94 ± 2	273 ± 64	6

PaO_2_ and SpO_2_ values: means ± STDs.

**Table 3 diagnostics-13-03312-t003:** PaO_2_ and oxemic-risk for each SpO_2_ target range.

	88–92%	89–93%	90–94%	91–95%	92–96%	93–97%
PaO_2_ (mmHg)	56 ± 24	59 ± 26	63 ± 31	67 ± 32	73 ± 37	80 ± 41
Oxemic-risk (95% CI)	1.24 (1.18–1.29)	1.12 (1.07–1.18)	0.95 (0.90–1.00)	0.83 (0.78–0.87)	0.80 (0.76–0.85)	0.88 (0.83–0.92)

PaO_2_ means ± STDs.

**Table 4 diagnostics-13-03312-t004:** PMA ranges and associated oxemic-risks for each SpO_2_ target range.

SpO_2_ Target Range	Overall	Low-PMA	Middle-PMA	High-PMA
88–92%				
PMA break		<39^5^	39^5^–42^1^	>42^1^
Oxemic-risk	1.24	1.26	1.38	1.02
89–93%				
PMA break		≤42^2^	*	>42^2^
Oxemic-risk	1.12	1.15		1.01
90–94%				
PMA break		≤42^3^	*	>42^3^
Oxemic-risk	0.95	1.02		0.67
91–95%				
PMA break		≤42^2^	*	>42^2^
Oxemic-risk	0.83	0.89		0.59
92–96%				
PMA break		≤34^3^	34^3^–42^3^	>42^3^
Oxemic-risk	0.80	0.74	0.89	0.62
93–97%				
PMA break		≤38^3^	38^4^–42^6^	>42^6^
Oxemic-risk	0.88	0.83	1.00	0.73

* The three center SpO_2_ target ranges are dichotomous. PMA weeks^days^.

## Data Availability

The data presented in this study are available on request from the corresponding author. The data are not publicly available due to privacy.

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
