# Peer review of "Determining an Optimal Oxygen Saturation Target Range Based on Neonatal Maturity: Demonstration of a Decision Tree Analytic"

_diagnostics, 2023, doi:10.3390/diagnostics13213312_

Round 1
Reviewer 1 Report
The authors present an interesting analysis of methods based on machine learning to minimize the risk of inappropriate oxygenation on preterm infants.
Only minor changes are here suggested to improve the quality of the paper:
1)Oxemic-risk: "Oxemic-risk" is not a standard term. It may require clarification or a definition.
2)SpO2, PaO2, and ICU: Ensure that these abbreviations are defined the first time they appear in the text, unless they are common knowledge in your field.
3) Sentence Fragment: "An observational study of 3 years of paired age-SpO2-PaO2 data from a neonatal ICU." This is not a complete sentence.
4) "Risk was based on a severity-weighted average of arterial oxygen outside the normal range for neonates (50-80 mmHg).": The sentence could be clearer.
5) Awkward Phrasing: "These results suggest that different SpO2 target ranges than is typical might reduce morbidity and mortality." could be better phrased.
Line 159: weeks/days incorrect
Author Response
We are glad that you enjoyed our article, and thank you for the recommendations regarding minor edits.
1)Oxemic-risk: "Oxemic-risk" is not a standard term. It may require clarification or a definition.
Thanks, we had overlooked this point. We have now made it clear in the METHODS that this term was chosen by us to address both the risk of hypoxemimia and hyperoxemia.
3) Sentence Fragment: "An observational study of 3 years of paired age-SpO2-PaO2 data from a neonatal ICU." This is not a complete sentence.
4) "Risk was based on a severity-weighted average of arterial oxygen outside the normal range for neonates (50-80 mmHg).": The sentence could be clearer.
5) Awkward Phrasing: "These results suggest that different SpO2 target ranges than is typical might reduce morbidity and mortality." could be better phrased.
We have rephrased these.
Line 159: weeks/days incorrect
We have amended the Methods section to define the presentation of the gestational age.
Reviewer 2 Report
Dear Editors,
Thank you for allowing me to review this fascinating article discussing the ideal oxygen saturation of neonatal maturity and artificial intelligence. I found it an engaging read, and I especially appreciated the authors' ability to synthesize complex ideas clearly and concisely, which can be challenging to come across in scientific literature. While I acknowledge that the study design raises some concerns about the validity of the conclusions, they provide a solid foundation for future research. Overall, I enjoyed reading this article and suggest a few minor revisions to improve its clarity and impact.
I recommend that the authors review the article's writing as it gives the impression that numerous double spaces make the text unpleasant.
The authors should take a more systematic approach in the methods section. Starting with a clear definition of the type of study, they should then describe the eligible population or characteristics of the center, data collection, and other relevant details before delving into the more specific aspects of the analysis. A well-described methodology can enhance the reproducibility and refutability of the work and can also help readers better understand any potential biases, such as selection bias. Although the authors note that some methodological aspects have been previously described (line 108), it would be helpful if they could provide more specific data. One possible way to achieve this would be to divide the methods section into subsections.
In methods, it would also be useful for the authors to describe some of the sections related to data analysis and, above all, ethical aspects. Although there is no direct interaction with the patients, there is the handling of clinical data collected from actual patients, so ethical aspects must be mentioned. Among them whether an ethics committee has evaluated the study.
In methods and results, it would be advisable always to use the simple past tense since sometimes, for example, the present tense is used.
The results section sometimes contains text with a slightly narrative tendency, for example, in the first phase. The authors could revise this to leave the sentences in this section as purely enunciative: an ideal results section is limited to describing what the authors obtain without narrating or assessing anything for the moment.
Although in the limitations section, the authors refer to the origin of their sample, a tertiary hospital, they do so to contextualize a particular aspect (which is relevant). However, it would be helpful if the authors described their research's potential and unavoidable selection bias. To this end, it would be helpful to describe the general characteristics of the center where the study is conducted and some characteristic(s) of the sample, which would allow the reader to contextualize that potential selection bias better. The authors could evaluate this bias, commenting on the implications for the external validity of their findings.
The English language used in the text is correct.
However, I recommend some general revisions to improve the overall writing quality. For instance, it would be better to avoid double spaces and maintain consistency in the past tense, especially in the methods and results sections. Additionally, it would be beneficial to eliminate any redundancies or overly explanatory statements in these sections.
Author Response
Thank you so much, we really appreciate that you found the manuscript so engaging. We have addressed your suggested edits, and they certainly enhance it further.
I recommend that the authors review the article's writing as it gives the impression that numerous double spaces make the text unpleasant.
We have corrected this.
The authors should take a more systematic approach in the methods section. Starting with a clear definition of the type of study, they should then describe the eligible population or characteristics of the center, data collection, and other relevant details before delving into the more specific aspects of the analysis. A well-described methodology can enhance the reproducibility and refutability of the work and can also help readers better understand any potential biases, such as selection bias. Although the authors note that some methodological aspects have been previously described (line 108), it would be helpful if they could provide more specific data. One possible way to achieve this would be to divide the methods section into subsections. In methods, it would also be useful for the authors to describe some of the sections related to data analysis and, above all, ethical aspects. Although there is no direct interaction with the patients, there is the handling of clinical data collected from actual patients, so ethical aspects must be mentioned. Among them whether an ethics committee has evaluated the study.
Your suggestion of structuring this section with subheads, was excellent, and we followed it. It further highlighted the need to fill in some of the gaps, such as ethics committee oversight, the data collection and analysis.
In methods and results, it would be advisable always to use the simple past tense since sometimes, for example, the present tense is used.
Thank you for spotting this problem. We edited both sections to correct the verb tense. We would note, however, that present tense might be appropriate on occasion. As an example when referring to a Table rather than the study results. e.g. the SpO2 values are shown in the Table vs. the SpO2 values showed. Most of the authors are native english speakers and the tenses just flow with the composition, though not always correctly.
The results section sometimes contains text with a slightly narrative tendency, for example, in the first phase. The authors could revise this to leave the sentences in this section as purely enunciative: an ideal results section is limited to describing what the authors obtain without narrating or assessing anything for the moment.
This is a relevant observation. We discussed it before the first submission. We decided to include these in consideration of the likely readers. While they would be inappropriate for the clinical audience, we thought they were helpful for the data scientists unfamiliar with neonatal intensive care. These were, by intent, only used for descriptive data not only so that the DISCUSSION could focus on the aim of the study, but also so that readers would not dwell on these issues irrelevant to the aim.
Although in the limitations section, the authors refer to the origin of their sample, a tertiary hospital, they do so to contextualize a particular aspect (which is relevant). However, it would be helpful if the authors described their research's potential and unavoidable selection bias. To this end, it would be helpful to describe the general characteristics of the center where the study is conducted and some characteristic(s) of the sample, which would allow the reader to contextualize that potential selection bias better. The authors could evaluate this bias, commenting on the implications for the external validity of their findings.
Excellent point, especially relating to the mixed readership. In the METHODS we have added detail to study population, and in the limitations highlighted the ability to project the results to other populations.
The English language used in the text is correct. However, I recommend some general revisions to improve the overall writing quality. For instance, it would be better to avoid double spaces and maintain consistency in the past tense, especially in the methods and results sections. Additionally, it would be beneficial to eliminate any redundancies or overly explanatory statements in these sections.
As noted above we have addressed these in our revision.